# Phylogenetic Relationships of the Strongyloid Nematodes of Australasian Marsupials Based on Mitochondrial Protein Sequences

**DOI:** 10.3390/ani12212900

**Published:** 2022-10-22

**Authors:** Tanapan Sukee, Ian Beveridge, Anson V. Koehler, Ross S. Hall, Robin B. Gasser, Abdul Jabbar

**Affiliations:** Department of Veterinary Biosciences, School of Veterinary Science, Faculty of Veterinary and Agricultural Sciences, The University of Melbourne, Werribee, VIC 3030, Australia

**Keywords:** Cloacininae, Phascolostrongylinae, Strongyloidea, mitochondrial protein-coding genes, phylogenetics

## Abstract

**Simple Summary:**

Parasitic strongyloid nematodes endemic to the gastrointestinal tracts of Australasian marsupials are one of the most diverse groups of mammalian parasites. These nematodes are currently placed in the family Chabertiidae comprising two subfamilies, namely the Cloacininae and Phascolostrongylinae. Their current classification relies primarily on morphological features and has not been validated using molecular data. This study aimed to determine the phylogenetic relationships of the Cloacininae and Phascolostrongylinae within the family Chabertiidae and their relationship with other groups of strongyloid nematodes from non-marsupial hosts, using mitochondrial protein sequence datasets. The findings supported the recognition of the family Cloacinidae, containing the Cloacininae and Phascolostrongylinae, as a monophyletic group within the Strongyloidea. However, the subfamily Phascolostrongylinae was paraphyletic, and the relationships of individual genera corresponded with their host families. Genera of the Cloacininae and Phascolostrongylinae occurring in macropod hosts were more closely related compared to genera of the Phascolostrongylinae occurring in wombats. This study suggests an alternative hypothesis for the origin of marsupial strongyloid nematodes in vombatid hosts that should be explored further using molecular approaches and more widespread sampling.

**Abstract:**

Australasian marsupials harbour a diverse group of gastrointestinal strongyloid nematodes. These nematodes are currently grouped into two subfamilies, namely the Cloacininae and Phascolostrongylinae. Based on morphological criteria, the Cloacininae and Phascolostrongylinae were defined as monophyletic and placed in the family Cloacinidae, but this has not been supported by molecular data and they are currently placed in the Chabertiidae. Although molecular data (internal transcribed spacers of the nuclear ribosomal RNA genes or mitochondrial protein-coding genes) have been used to verify morphological classifications within the Cloacininae and Phascolostrongylinae, the phylogenetic relationships between the subfamilies have not been rigorously tested. This study determined the phylogenetic relationships of the subfamilies Cloacininae and Phascolostrongylinae using amino acid sequences conceptually translated from the twelve concatenated mitochondrial protein-coding genes. The findings demonstrated that the Cloacininae and Phascolostrongylinae formed a well-supported monophyletic assemblage, consistent with their morphological classification as an independent family, Cloacinidae. Unexpectedly, however, the subfamily Phascolostrongylinae was split into two groups comprising the genera from macropodid hosts (kangaroos and wallabies) and those from vombatid hosts (wombats). Genera of the Cloacininae and Phascolostrongylinae occurring in macropodid hosts were more closely related compared to genera of the Phascolostrongylinae occurring in wombats that formed a sister relationship with the remaining genera from macropods. These findings provide molecular evidence supporting the monophyly of the family Cloacinidae and an alternative hypothesis for the origin of marsupial strongyloid nematodes in vombatid hosts that requires further exploration using molecular approaches and additional samples

## 1. Introduction

Australasian marsupials are hosts to a diverse and abundant range of nematodes belonging to the superfamily Strongyloidea. Currently, over 300 species have been described from the gastrointestinal tracts of herbivorous marsupials that include the Macropodoidea (kangaroos, wallabies, rat-kangaroos and potoroos) and Vombatidae (wombats) [1]. They are currently divided into two subfamilies, the Cloacininae and the Phascolostrongylinae based on morphological features. However, the phylogenetic relationships between these two subfamilies have not been rigorously tested using molecular data.

The subfamily Cloacininae is found exclusively in the stomachs or oesophagi of kangaroos, wallabies, potoroos and rat-kangaroos (families Macropodidae, Potoroidae and Hypsiprymnodontidae) while species of the Phascolostrongylinae are found in the stomachs or large intestines of macropodoids (families Macropodidae and Hypsiprymnodontidae) as well as in the colon of wombats (family Vombatidae) [2].

Morphologically, the subfamily Cloacininae is defined by a cylindrical buccal capsule, the externo-dorsal ray of the male bursa arising from the lateral trunk and usually a Type II or J-shaped female ovejector. This subfamily has been subdivided into six tribes based on the morphology of their oral and buccal features as well as the oesophagus [3]. The subfamily Phascolostrongylinae consists of three tribes and is characterised by cylindrical or globular buccal capsules with a dorsal gutter, the externo-dorsal ray arising from the dorsal trunk and mostly a Type I or Y-shaped ovejector [2].

The subfamily Cloacininae has recently been subjected to extensive morphological revision [3]. However, its molecular phylogeny based on the first and second internal transcribed spacers (ITS) of the ribosomal DNA does not entirely support the morphological classification and requires further validation [4]. As for the subfamily Phascolostrongylinae, extensive molecular studies carried out using both the ITS and mitochondrial amino acid sequence data found that phylogenetic relationships within this group were correlated with their host families [5,6]. However, the phylogenetic relationships between the subfamilies Phascolostrongylinae and Cloacininae remain uncertain. Based on morphological data, the Cloacininae and Phascolostrongylinae were hypothesised to be monophyletic and placed together within the family Cloacinidae by Beveridge [2]. Although this hypothesis was tested using the ITS-2 sequence data, the results were inconclusive due to inadequate sample size and statistical support for the phylogenetic analyses [7].

Mitochondrial protein sequences have been used previously to validate phylogenetic hypotheses of phascolostrongyline nematodes with improved statistical support [6,8,9]. This study aimed to characterise the mitochondrial protein-coding sequences of representatives of the subfamily Cloacininae using next-generation sequencing in order to infer the phylogenetic relationships between the subfamilies Cloacininae and Phascolostrongylinae as well as their position within the superfamily Strongyloidea.

## 2. Materials and Methods

### 2.1. Sample Collection and DNA Extraction

Adult female nematodes belonging to five species representing the subfamily Cloacininae (Table 1) were obtained from the frozen parasite collection at the School of Veterinary Science, The University of Melbourne. These specimens had been collected from the gastrointestinal tracts of carcasses of hosts from commercial culls or vehicle collisions. The samples had been either frozen at −80 °C or preserved in 70% ethanol and then frozen at −80 °C as individuals or pools. For morphological identification, the anterior and posterior extremities of each nematode were removed with a scalpel blade, cleared with lactophenol and identified. They were subsequently deposited in the Australian Helminthological Collection (AHC) of the South Australia Museum, Adelaide, as voucher specimens under the registration numbers AHC 49109–13. The mid-sections were used for DNA extraction.

Genomic DNA was extracted from individual nematodes using the QiaAmp Micro kit (Qiagen, Hilden, Germany) following the manufacturer’s protocol for extracting DNA from tissues. For molecular identification, the ITS-1 and ITS-2 sequences were determined for each individual worm using an established PCR-based sequencing method [6]. Prior to the next-generation sequencing, the quantity and quality of the DNA were assessed using the 2200 TapeStation (Agilent, Santa Clara, CA, USA).

### 2.2. Sequencing and Gene Annotation

Illumina TruSeq indexed libraries were prepared using sheared DNA following the manufacturer’s protocol (Illumina, San Diego, CA, USA). Briefly, the steps included (i) end-repair and A-tailing of the 3′ ends, (ii) ligation of the adaptors, (iii) enrichment of the libraries and purification of the enriched library using Ampure Beads (Beckman Coulter, Brea, CA, USA). The libraries were quantified using the 2200 TapeStation, pooled and sequenced on the Illumina MiSeq platform using the 300 cycle v3 reagent kit (2 × 150 paired-end reads).

Raw sequence data in the FASTQ format were filtered for quality in Trimmomatic v.0.38 [15] prior to de novo assembly, employing the program Spades v3.13.0 under default parameters. For assembly, each of the 12 protein-coding genes of the mitochondrial genome were identified by local sequence alignment (six reading frames) using the amino acid sequence inferred from corresponding genes of reference mitochondrial genomes using an established workflow [11]. The mitochondrial genome sequence of *Hypodontus macropi* (NC023083) was used as a reference [8]. The nucleotide sequences of the 12 protein-coding genes of each species included in this study were deposited in the GenBank database under the accession numbers OL582565–OL582569.

### 2.3. Sequence Comparison and Phylogenetic Analyses

The nucleotide sequences of the 12 mitochondrial protein-coding genes and the encoded proteins were aligned separately using CLUSTAL W [16] and MUSCLE [17], followed by a concatenation of the alignments using MEGA v.10 [18]. Published mitochondrial protein-coding gene sequences of the Phascolostrongylinae, Chabertiinae, Oesophagostominae, Strongylidae and Syngamidae were included in the alignments (Table 1). Pairwise comparisons of the nucleotide and amino acid sequences were calculated using MEGA. The nucleotide diversity (Pi) of the mitochondrial protein-coding genes was determined using the Sliding Window Analysis (SWAN) conducted in the program DnaSP v.5 [19] with a sliding window of 100 base pairs (bp) and step size of 25 (bp).

Bayesian Inference (BI) phylogenetic analysis was conducted in MrBayes v.3.2.7 [20] using concatenated, aligned amino acid sequences derived from all 12 mitochondrial genes. *Syngamus trachea* (GQ888718), belonging to the family Syngamidae, was included as the outgroup. The optimal partitioning schemes and substitution model were determined using PartitionFinder 2 [21] for amino acids, with a model selection set to Akaike Information Criterion and greedy search. According to PartitionFinder, the concatenated alignment was partitioned into the following eight subsets: subset 1 (*cox*1 and *cox*2), subset 2 (*nad*3 and *nad*4), subset 3 (*nad*5), subset 4 (*nad*6 and *nad*2), subset 5 (*nad*1 and *nad*4L), subset 6 (*atp*6), subset 7 (*cytb*) and subset 8 (*cox*3). The BI analysis was conducted with four chains for 5 million Markov Chain Monte Carlo iterations, sampling every 1000th generation for four independent runs. The evolutionary model set for subset 6 was mtrev while mtmam was applied to the remaining subsets. Convergence was determined by the average standard deviation of split frequencies of >0.01 and the potential scale reduction factor approaching 1. A proportion (25%) of the sampled trees was discarded as burnin and the consensus tree was constructed from the remaining 75%. The consensus tree was visualised in FigTree v.1.4.4 [22].

## 3. Results

### 3.1. Nucleotide and Amino Acid Sequence Comparisons

The 12 concatenated mitochondrial protein-coding genes were aligned over 10,380 bp, and respective protein sequences over 3440 amino acids (aa). Within the subfamily Cloacininae, pairwise sequence difference ranged between 5.4–8.4% (aa) and 14.1–16.6% (nt) (Table 2). *Macroponema* and *Rugopharynx* shared the greatest nucleotide sequence similarity whereas *Parazoniolaimus* and *Zoniolaimus* shared the greatest amino acid sequence similarity. The genera *Cloacina* and *Zoniolaimus* shared the least similarity in both amino acid and nucleotide sequences.

Pairwise sequence differences between the subfamilies Cloacininae and Phascolostrongylinae ranged from 7.6–11.1% (aa) and 14.1–16.6% (nt). The nucleotide and amino acid sequences of *Rugopharynx* from the subfamily Cloacininae were most similar to *Paramacropostrongylus* (*P. typicus*) from the subfamily Phascolostrongylinae. The least similar sequences (both aa and nt) were between those of *Macropostrongyloides baylisi* and *Cloacina communis*. There was a greater amino acid sequence similarity between the cloacinine genera and the phascolostrongyline genera from wombats (7.6–9.4%) compared to those from macropodid host (7.9–10.6%), with the exception of *Macropostrongyloides phascolomys* from the common wombat, *Vombatus ursinus* (9–10.8%).

Nucleotide diversity (Pi) across the alignment of 12 concatenated mitochondrial protein-coding genes of all sequences included in the phylogenetic analyses ranged from the 0.093 (*cox*1) to 0.28 (*nad*5) (Figure 1).

### 3.2. Phylogenetic Analyses

Bayesian Inference analysis of the 12 concatenated amino acid sequences of the mitochondrial genes of 35 taxa of the Strongyloidea resulted in six clades (Figure 2). The five genera representing the subfamily Cloacininae from the stomachs of macropodid marsupials formed a clade with a posterior probability (pp) of 1.00, while the genera of the subfamily Phascolostrongylinae were divided between two clades. One of these clades comprised the phascolostrongyline genera from the stomachs and intestines of macropodids (*Macropostrongyloides phascolomys* from the common wombats was the exception in this clade) (pp = 1.00) and had a sister relationship with the clade containing the cloacinine genera (pp = 1.00). The other phascolostrongyline clade consisted of *Oesophagostomoides* spp. and *Phascolostrongylus turleyi* from the large intestines of vombatid marsupials (pp = 1.00). This clade was sister to both the cloacinine clade and the clade comprising the phascolostrongyline genera from macropodids (pp = 0.99).

The topology within the clade containing the subfamily Cloacininae showed that *Zoniolaimus dendrolagi* and *Parazoniolaimus collaris* formed a monophyletic group (pp = 1.00), sister to *Macroponema beveridgei*, with strong nodal support (pp = 1.00). *Cloacina communi*s was placed on a branch sister to the remaining genera of Cloacininae included in the analysis (pp = 1.00).

There were two poorly supported nodes within the clade comprising the Phascolostrongylinae from macropodid marsupials. These were the sister relationship between *Torquenema toraliforme* and *Macropostrongyloides* spp. (pp = 0.71) with *Wallabicola dissimilis* on a branch external to *Hypodontus*, *Macropicola* and *Macropostrongyloides* spp. (pp = 0.88). Low nodal support was encountered in the clade consisting of *Phascolostrongylus* and *Oesophagostomoides* spp. found exclusively in wombats. There was a low posterior probability (0.84) of *Phascolostrongylus turleyi* associating with *Oesophagostomoides longispicularis* and the sister relationship between these species and *Oe. stirtoni* was also poorly supported (pp = 0.84). The BI analyses showed that the tribe Oesophagostominea was paraphyletic. *Oesophagostomum dentatum* and *Oe. quadrispinulatum* formed a well-supported clade sister to the strongyloids from marsupials (pp = 1.00). *Oesophagostmum asperum* was sister to the clade comprising the Chabertiinea (*Chabertia* spp.), while *Oesophagostomum columbianum* was placed on a branch external to the remaining members of the Chabertiinae and Oesophagostiminae included in the analysis (pp = 1.00).

## 4. Discussion

The phylogenetic relationship between the subfamilies Cloacininae and Phascolostrongylinae determined by BI analysis of the mitochondrial protein sequences provides the first molecular support for the monophyly of the family Cloacinidae, an association previously supported primarily by host associations. However, the paraphyly of the subfamily Phascolostrongylinae was unexpected and conflicted with the current morphological classification. The current data suggest that the Phascolostrongylinae needs to be split into two subfamilies with the genera from wombats remaining in the Phascolostrongylinae and a new subfamily created for the genera found in macropodids.

In the current study, the subfamily Phascolostrongylinae was divided into two groups comprising the genera exclusive to wombats and the genera found in macropodoids, with the latter group being more closely related to the Cloacininae. The paraphyly of the Phascolostrongylinae in this study contrasts with the finding from a previous phylogenetic study [5] due to the inclusion of the five genera from the subfamily Cloacininae. This relationship does not support the morphological classification of the Cloacininae and Phascolostrongylinae as two distinct monophyletic subfamilies [2]. The Cloacininae and Phascolostrongylinae are currently differentiated primarily by the origin of the externo-dorsal rays of the bursa, the presence or absence of a dorsal gutter and the shape of the ovejector [2,23,24]. However, these features, apart from the dorsal gutter, do not consistently differentiate the Cloacininae from the Phascolostrongylinae. For instance, *Wallabicola dissimilis* from the subfamily Phascolostrongylinae possesses a J-shaped ovejector [25], which is characteristic of the Cloacininae, whereas the remaining genera of the Phascolostrongylinae possess a Y-shaped ovejector, or a slight variation to it [2]. The externo-dorsal rays usually originate from the dorsal trunk in the Phascolostrongylinae and from the lateral trunk in the Cloacininae. There are, however, two species of *Cloacina* (Cloacininae) from the banded-hare wallaby, *Lagostrophus fasciatus,* in which the externo-dorsal ray arises from the dorsal ray [26]. Recently, Sukee et al. [25] pointed out that the shape of the deirid is a reliable character for differentiating the Cloacininae from the Phascolostrongylinae since it is consistently elongate and setiform in the former compared to short and papillate in the latter. Although this feature could be considered taxonomically important and used in future, additional morphological features would facilitate the differentiation between the Cloacininae and the Phascolostrongylinae.

The current tree placed the phascolostrongyline genera exclusive to vombatid marsupials as the sister group of the remaining genera included in the study. This finding is inconsistent with previous hypotheses on the radiation of the Strongyloidea in marsupials which was thought to have originated primarily in the Macropodidae [27]. The genera *Hypodontus*, *Macropicola* and *Corollostrongylus* were proposed as ancestral to the remaining genera because of their globular buccal capsules and Y-shaped ovejectors, considered to be plesiomorphic characters [2]. The phascolostrongyline genera from wombats with cylindrical buccal capsules, a shared apomorphy with the Cloacininae, were thought to have arisen by means of host-switching [2]. Although this assumption conflicts with the current phylogenetic tree, host-switching did occur in the case of *Macropostrongyloides phascolomys* and explains its position as the only wombat-inhabiting species nested among the phascolostrongyline genera from macropodids.

The position of *Phascolostrongylus* and *Oesophagostomoides* as a sister-group to the genera in macropodids suggests the possibility of the strongyloids of marsupials originating from representatives in the Vombatidae, a hypothesis considered by Beveridge [28]. This is plausible, given the evolution of marsupials, whereby the ancestors of the surviving wombats, the Vombatiformes, were present in Australia during the late Oligocene [29] which is prior to the radiation of the Macropodidae [30]. The only other molecular comparison between parasitic helminths of macropodid and vombatid marsupials is that of Hardman et al. [31], in which the anoplocephalid cestode genera *Phascolotenia* and *Phascolocestus* (formerly *Paramoniezia*) from wombats formed a sister clade to genera from macropodids. However, the genera *Vombatus* and *Lasiorhinus* are the only surviving species remaining of the formerly extensive vombatiform radiation [29,32]. Therefore, it would be difficult to properly test this hypothesis unless nematodes can be recovered from fossilised material such as coprolites as have been achieved with the strongylid parasites of the extinct moas of New Zealand [33].

The phylogenetic relationship between the Cloacininae and Phascolostrongylinae was related both to host families and predilection sites within hosts. Strongyloid nematodes inhabit sections of the gastrointestinal tract in which fermentative digestion occurs, which is primarily the large intestine and, in the case of macropodids, both the stomach and the large intestine [34]. For instance, the genera *Strongylus*, *Cylicodontophorus*, *Chabertia, Oesophagostomum* are all parasites of the large intestines of equids, rodents, primates, pigs and ruminants [24]. However, in the Phascolostrongylinae there is a split between genera occurring in the large intestines and the stomach, while the genera of the Cloacininae are found mostly in the stomachs with a few genera in the oesophagus. The wombats and early macropodoids such as the musky-rat kangaroo, *Hypsiprymnodon moschatus,* are monogastric and the large intestine is the only site in which fermentative digestion occurs [34]. *Corollostrongylus hypsiprymnodontis* is the only strongyloid species found in the musky-rat kangaroo; however, specimens were not available for molecular studies.

In this study, BI analyses of the concatenated mitochondrial protein sequences resulted in a phylogenetic tree with mostly strong nodal support. Bayesian Inference was chosen over the conventional neighbour joining, distance-based method due to its greater accuracy for estimating distant relationships [35]. In addition, BI is more robust for the phylogenetic analysis of protein datasets, particularly when a gamma distribution is used to accommodate variations of rates among sites [36]. Sliding window analysis of the 12 concatenated mitochondrial protein-coding genes showed results consistent with previous studies, indicating that some genes with high nucleotide diversity, such as *nad*1 and *nad*5, might be utilised as complementary markers to ITS-1 and ITS-2 [5,8].

## 5. Conclusions

This study provided the first molecular evidence to show that the subfamilies Cloacininae and Phascolostrongylinae are a monophyletic group, valid under the family Cloacinidae and separated from the strongyloids from non-marsupial hosts. Furthermore, this study showed that the subfamily Phascolostrongylinae is paraphyletic and warrants future taxonomic revisions. Although additional studies are required, current molecular evidence supports the hypothesis that early radiation of the strongyloid genera in marsupials may have occurred in the Vombatidae. Future studies could utilise the mitochondrial protein sequence data sets for deeper investigations of the evolution of strongyloid nematodes.

## Figures and Tables

**Figure 1 animals-12-02900-f001:**
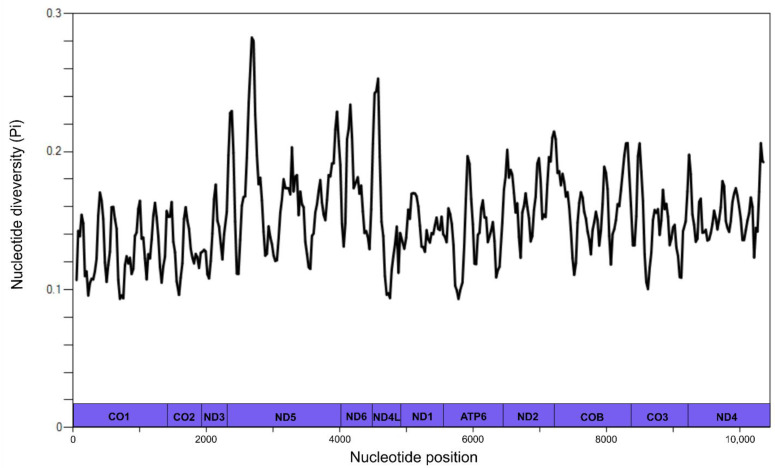
Nucleotide diversity (Pi) across twelve concatenated mitochondrial protein-coding genes (y-axis) of the Cloacininae, Phascolostrongylinae, Strongylidae, Oesophagostominae, Chabertiinae and Syngamidae included in the alignment. Nucleotide diversity was calculated in the software DnaSP version 6 using a window of 100 bp and 25 bp-steps. The nucleotide position (base pairs) is indicated on the X-axis next to the boundaries between mitochondrial protein-coding genes.

**Figure 2 animals-12-02900-f002:**
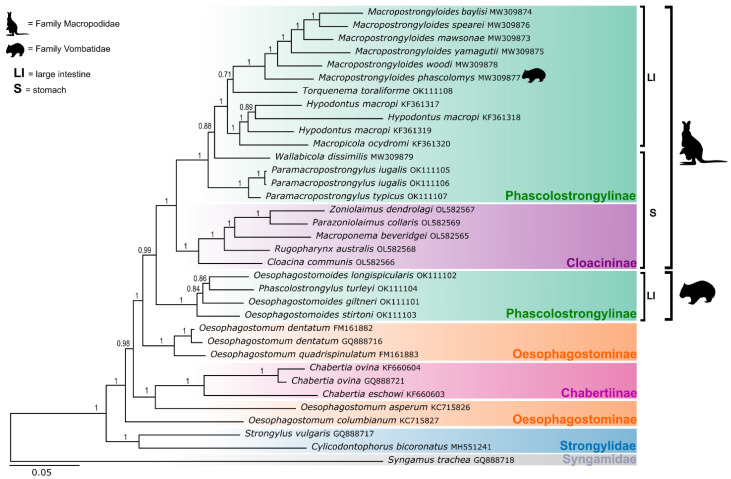
Topology of the Bayesian Inference phylogenetic analyses inferred from the concatenated alignment of the amino acid sequence of 12 mitochondrial genes of the genera of Cloacininae, Phascolostrongylinae, Oesophagostominae and Chabertiinae (Chabertiidae) and Strongylidae. Nodal support is indicated as posterior probabilities of the Bayesian Inference analysis. *Syngamus trachea* from the family Syngamidae was used as the outgroup. The host families (Macropodidae or Vombatidae) in which the species of Cloacininae and Phascolostrongylinae occur are represented by icons. The scale bar indicates the number of inferred substitutions per site.

**Table 1 animals-12-02900-t001:** Representatives of the subfamilies Cloacininae, Phascolostrongylinae, Chabertiinae, Oesophagostominae, families Strongylinae and Syngaminae included in the phylogenetic analyses.

Family or Subfamily	Species	Host Species	Collection Locality	GenBank Accession Mo.	Reference
Subfamily Cloacininae	*Cloacina communis*	*Osphranter robustus*	Menzies, WA, Australia	OL582566	this study
	*Macroponema beveridgei*	*Osphranter robustus*	Hillgrove Stn, Charters Towers, Qld, Australia	OL582565	this study
	*Parazoniolaimus collaris*	*Wallabia bicolor*	The Gurdies, Vic, Australia	OL582569	this study
	*Rugopharynx australis*	*Osphranter rufus*	Wallerberdina Stn, Port Augusta, SA, Australia	OL582568	this study
	*Zoniolaimus dendrolagi*	*Dendrolagus lumholtzi*	East Beatrice, Qld, Australia	OL582567	this study
Subfamily Phascolostrongylinae	*Oesophagostomoides giltneri*	*Vombatus ursinus*	Flowerdale, Vic, Australia	OK111101	[5]
	*Oesophagostomoides longispicularis*	*Vombatus ursinus*	Gippsland, Vic, Australia	OK111102	[5]
	*Oesophagostomoides stirtoni*	*Lasiorhinus latifrons*	Swan Reach, SA, Australia	OK111103	[5]
	*Paramacropostrongylus iugalis*	*Macropus giganteus*	Miles, Qld, Australia	OK111105	[5]
	*Paramacropostrongylus iugalis*	*Macropus giganteus*	Charters Towers, Qld, Australia	OK111106	[5]
	*Paramacropostrongylus typicus*	*Macropus fuliginosus*	Nyngan, NSW, Australia	OK111107	[5]
	*Phascolostrongylus turleyi*	*Vombatus ursinus*	Flowerdale, Vic, Australia	OK111104	[5]
	*Torquenema toraliforme*	*Macropus giganteus*	Research, Vic, Australia	OK111108	[5]
	*Macropostrongyloides mawsonae*	*Macropus giganteus*	Heathcote, Vic, Australia	MW309873	[9]
	*Macropostrongyloides baylisi*	*Osphranter robustus*	Cloncurry, Qld, Australia	MW309874	[9]
	*Macropostrongyloides yamagutii*	*Macropus fuliginosus*	Hattah Lakes, Vic, Australia	MW309875	[9]
	*Macropostrongyloides spearei*	*Osphranter robustus*	Kalgoorlie, WA, Australia	MW309876	[9]
	*Macropostrongyloides phascolomys*	*Vombatus ursinus*	Flowerdale, Vic, Australia	MW309877	[9]
	*Macropostrongyloides woodi*	*Osphranter rufus*	Kalgoorlie, WA, Australia	MW309878	[9]
	*Wallabicola dissimilis*	*Wallabia bicolor*	Kamarooka, Vic, Australia	MW309879	[9]
	*Hypodontus macropi*	*Wallabia bicolor*	Hall’s Gap, Vic, Australia	KF361317	[8]
	*Hypodontus macropi*	*Thylogale billardierii*	Launceston, Tas, Australia	KF361318	[8]
	*Hypodontus macropi*	*Macropus robustus*	Barcaldine, Qld, Australia	KF361319	[8]
	*Macropicola ocydromi*	*Macropus fuliginosus*	Waroona, WA, Australia	KF361320	[8]
Subfamily Oesophagostominae	*Oesophagostomum dentatum*	*Sus scrofa domestica*	Chongqing, China	FM161882	[10]
	*Oesophagostomum quadrispinulatum*	*Sus scrofa domestica*	Chongqing, China	FM161883	[10]
	*Oesophagostomum dentatum*	*Sus scrofa domestica*	Werribee, Vic, Australia	GQ888716	[11]
	*Oesophagostomum asperum*	*Capra hircus*	Shaanxi Province, China	KC715826	[12]
	*Oesophagostomum columbianum*	*Ovis aries*	Heilongjiang Province, China	KC715827	[12]
Subfamily Chabertiinae	*Chabertia ovina*	*Ovis aries*	Werribee, Vic, Australia	GQ888721	[11]
	*Chabertia ovina*	*Capra hircus*	Shaanxi Province, China	KF660604	[13]
	*Chabertia ershowi*	*Bos grunniens*	Qinghai Province, China	KF660603	[13]
Family Strongylidae	*Cylicodontophorus bicoronatus*	*Equus caballus*	Heilongjiang Province, China	MH551241	[14]
	*Strongylus vulgaris*	*Equus caballus*	Werribee, Vic, Australia	GQ888717	[11]
Family Syngamidae	*Syngamus trachea*	*Gymnorhina tibicen*	Werribee, Vic, Australia	GQ888718	[11]

Abbreviations: NSW = New South Wales, Qld = Queensland, SA = South Australia, Tas = Tasmania, WA = Western Australia, Vic = Victoria, Stn = Station.

**Table 2 animals-12-02900-t002:** Pairwise comparisons of the amino acid (bottom left) and nucleotide (top right) sequences from representatives of the subfamilies Cloacininae, Phascolostrongylinae, Oesophagostominae and Chabertiinae.

	1	2	3	4	5	6	7	8	9	10	11	12	13	14	15	16	17	18
1. *C. communis* 35T2		14.3	14.2	13.9	15.4	15.8	15.2	15.1	15.0	14.9	14.3	15.3	14.5	14.7	14.8	15.0	14.9	15.9
2. *Mac. beveridgei* 21H1	7.9		13.6	12.8	14.4	15.8	15.5	15.7	15.3	14.9	14.1	15.6	14.8	15.5	15.4	15.3	15.0	16.0
3. *Pa. collaris* YE8	8.3	7.7		12.9	13.1	15.7	15.4	15.4	15.4	14.6	14.4	15.6	14.9	15.2	15.4	15.3	14.9	15.7
4. *R. australis* AC5	6.5	6.7	6.2		14.2	15.4	14.7	14.9	14.8	14.0	14.3	15.1	14.3	15.2	15.0	15.0	14.4	15.7
5. *Z. dendrolagi* 48Z2	8.4	7.8	5.4	7.2		16.6	16.2	15.8	15.9	15.3	15.3	16.5	15.8	15.9	16.4	15.9	16.1	16.9
6. *Ma. baylisi* 21v1	9.7	10.6	10.7	9.5	11.1		15.1	15.0	15.1	14.2	14.5	13.7	16.2	16.1	16.0	16.3	15.8	16.7
7. *H. macropi* KF361317	8.4	8.9	9.6	8.2	9.8	8.4		13.6	14.8	13.8	14.0	14.2	15.3	15.6	15.8	15.9	15.3	16.0
8. *M. ocydromi* KF361320	8.8	9.9	10.0	8.9	10.3	8.3	6.5		14.4	13.6	14.2	14.2	15.4	15.3	16.1	15.7	15.4	16.2
9. *W. dissimilis* 10W9	7.9	8.8	9.5	8.2	9.5	8.5	6.5	7.2		13.6	13.3	14.3	15.2	15.3	15.8	15.5	14.9	16.0
10. *T. toraliforme* YD5	8.4	9.5	9.5	8.3	10.1	7.9	6.9	7.1	6.6		12.9	13.5	14.7	15.4	15.1	15.2	14.7	15.8
11. *P. typicus* 14B28	7.8	8.9	8.8	7.6	9.2	8.2	6.7	7.1	5.8	6.4		13.9	14.6	14.7	14.8	14.4	14.4	15.7
12. *Ma. phascolomys* 41R1	9.1	10.5	10.4	9.3	10.8	7.0	7.5	7.5	7.4	7.2	7.4		15.7	15.6	15.8	15.8	15.6	16.7
13. *O. stirtoni* 41W1	7.7	9.2	9.1	8.1	9.4	10.1	8.2	9.2	8.3	8.2	7.8	9.2		12.6	12.5	12.4	14.3	15.5
14. *O. giltneri* 41Z1	8.3	9.5	9.4	8.6	9.7	10.1	8.5	9.4	8.4	8.8	8.0	9.7	4.9		12.4	12.4	14.5	16.2
15. *Ph. turleyi* 42L2	8.2	9.9	9.9	8.6	9.9	10.4	9.3	9.8	8.8	9.0	8.4	9.8	4.7	4.9		12.4	14.5	16.1
16. *O. longispicularis* 47F	7.85	9.36	9.39	8.17	9.42	10.1	8.3	9.5	8.3	8.3	7.9	9.4	4.9	4.8	4.9		14.5	16.2
17. *Oe. dentatum* FM161882	7.59	8.6	8.08	7.88	8.95	9.91	8.1	8.9	7.9	8.3	7.7	9.2	6.4	6.7	7.0	6.6		15.6
18. *Ch. ovina* GQ888721	11.2	12	11.9	11.4	12.2	13	11.3	11.8	11.2	11.5	11.2	12.1	10.4	10.6	11.3	10.9	9.5	

## Data Availability

Specimens were deposited in the Australian Helminthological Collection (AHC) of the South Australian Museum, Adelaide (SAM), Australia. The mitochondrial DNA sequences of reported in this manuscript are available from the GenBank database.

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
