# Peer review of "Phylogenetic Relationships of the Strongyloid Nematodes of Australasian Marsupials Based on Mitochondrial Protein Sequences"

_animals, 2022, doi:10.3390/ani12212900_

Round 1
Reviewer 1 Report
The MS animals-1910658 describes the relation between the parasitic strongyloids of marsupials based on sequences of about 3400 amino acids, coded by the mitochondrial genome. The relation between the different strongyloids is important in the evolution of these worms and in the evolution of parasitism in general. The MS is well written and the conclusions are logical. I have only 1 remark.
I tried to confirm the cladistics by using maximum likelihood (MEGA X). The sequences of ref 9, were according Genbank unverified and therefore not included into my analysis. See below. The overall tree is similar to that in the MS and the Phascolostrongylinae were divided into 3 clades (LI from Macropodidae, Stomach from Macropodidae and LI from Vombatidae), with large bootstrap values, as in the MS. Therefore it is very well possible that the Cloacininae are situated in between. However, that cannot be checked, because the Cloacininae sequences were not available to me.
Reviewer 2 Report
The paper is in fact the result of adding five additional sequences to the ones described in the paper published by the same authors in 2021:
Sukee, T.; Beveridge, I.; Koehler, A.V.; Hall, R.; Gasser, R.B.; Jabbar, A. Phylogenetic relationships of the nematode subfamily Phascolostrongylinae from macropodid and vombatid marsupials inferred using mitochondrial protein sequence data. Parasit Vectors 2021, 14.
The conclusions, presented in the 2021 paper included the existence of two different branches in the Phascolostrongylinae subfamily. The main difference is the inclusion of five sequences from species of the subfamily Cloacininae. These five species occupy a distinct branch in an intermediate position between the two branches of the Phascolostrongylinae subfamily.
Even the position in the tree of Macropostrongyloides phascolomys was clearly indicated in the Parasit Vector paper in 2021. Also in 2021, the morphological description of this species was published in Animals by the same authors (doi https://doi.org/10.3390/ani11010175). In the conclusions, the phrase “Future studies utilising the nuclear and mitochondrial DNA sequence data could provide better insights into the evolution of this highly diverse group of parasitic nematodes from herbivorous Australian marsupials.” was included. Well, this future was quite close, since the paper based up in mitochondrial sequences was published by the same authors in the same year 2021 in Parasit Vectors.
I have no objections on the methodology and the conclusions of the paper (apart from some comments I have added to the pdf). However, I can’t help remarking that the amount of really new information available in this manuscript is very limited when compared to the paper in Parasit Vectors, and it derives from only five more mitochondrial sequences. The conclusions in the present manuscript are not discordant with the ones presented in 2021 for the rest of subfamilies and genera, but a small addition to the previous report and as a consequence:
-The title should reflect that the new data are referred only to Cloacininae.
-In my opinion, this amount of new information does not justify a Full Paper. A short note or a brief report should be more adequate formats. This is why I suggest major revision.
Reviewer 3 Report
This manuscript provides substantial new mitochondrial sequence data for an interesting group of nematodes that have been the focus of morphological and some (nuclear rDNA) studies. The phylogenetic analyses have been performed correctly, although one should be added. The manuscript is also well written. My comments provide a few insights that I think will improve the manuscript. Comment 4 is particularly important as this analysis would add to the conclusions.
1. The introduction discusses the morphological basis for the subfamilies. The description suggests that there is both overlap (buccal capsule) and variation in ovejector morphology that makes these characters inconsistent with the taxonomic groups. Is this a case where the first studies showed what appeared to be a clearcut distinction, but examination of more species showed exceptions? Perhaps a little more here about the strength of the morphological basis for monophyly and how strong it is supported? There is a bit more about this in the discussion, but a little more in the Introduction would be useful.
2. Some results from Table 2 are discussed in the text, but I am not certain I see the value of including Table 2. Of course the manuscript is otherwise quite concise, so it is not like it needs to be cut out because of length. I don't particularly see the value in pairwise distances in what is a phylogenetic paper.
3. Although few vombatid nematodes have been sequenced (Figure 2), the authors never seem to suggest that a future possibility is that a group of such parasites (most Phascolostrongylinae of vombatids) might be recognized as a separate subfamily. Maybe they don't want to raise this at this point; others surely would.
4. The authors discuss a bit about which strongyloids of marsupials might be ancestral. Earlier work is referenced. However, the authors don't make use of their own phylogeny to examine this hypothesis in more detail. The are many methods, including Bayesian methods that the authors seem to prefer that permit developing hypothesis from a phylogeny about which (in this case) host group is ancestral. The question becomes which host group (Macropodidae or Vombatidae) would be mapped to the node with 0.99 posterior probability in the tree. This should be determined and presented as an hypothesis for further testing as more species become available for phylogenetic analysis.
5. line 280 states "For instance, the genera Strongylus, Cylicodontophorus, Chabertia, Oesophagostomum are all parasites of the large intestines of equids and ruminants [24]."
My recollection is that Oesophagostomum is also found in primates and some other hosts not mentioned here.
6. line 299, "This study provided the first molecular evidence to show the monophyly of the subfamilies Cloacininae and Phascolostrongylinae with respect to the remaining strongyloid genera included and ..."
I understand that the authors are referring to the Cloacinae + Phascolostrongylinae are together monophyletic, but I think not all readers will understand this. After all, the authors showed that Phascolostrongylinae is not monophyletic. However, Cloacinidae is monophyletic in this study. My point is that the authors might be more careful in word choice for describing what is monophyletic.
Round 2
Reviewer 2 Report
In my previous report I wrote:
The conclusions, presented in the 2021 paper included the existence of two different branches in the Phascolostrongylinae subfamily. The main difference is the inclusion of five sequences from species of the subfamily Cloacininae. These five species occupy a distinct branch in an intermediate position between the two branches of the Phascolostrongylinae subfamily.
Even the position in the tree of Macropostrongyloides phascolomys was clearly indicated in the Parasit Vector paper in 2021.
In the V2 version, even if some modifications are made, no mention to the 2021 reprot are made. In this way, the readers may suppose this information on Phascolostrongylinae is new. As I told before, the new data are the new five sequences from Cloacininae.
Unless this difference among new and previous data is clearly stated, the redaction is unfair, and I can not agree with it.

Round 3
Reviewer 2 Report
No more comments